# Cloud-droplet growth due to supersaturation fluctuations in stratiform clouds

Xiang-Yu Li[1,2,3,4,5], Gunilla Svensson[1,3,6], Axel Brandenburg[2,4,5,7], and Nils E. L. Haugen[8,9]

[1]Department of Meteorology and Bolin Centre for Climate Research, Stockholm University, Stockholm, Sweden
[2]Nordita, KTH Royal Institute of Technology and Stockholm University, 10691 Stockholm, Sweden
[3]Swedish e-Science Research Centre, www.e-science.se, Stockholm, Sweden
[4]Laboratory for Atmospheric and Space Physics, University of Colorado, Boulder, CO 80303, USA
[5]JILA, Box 440, University of Colorado, Boulder, CO 80303, USA
[6]Global & Climate Dynamics, National Center for Atmospheric Research, Boulder, CO 80305, USA
[7]Department of Astronomy, Stockholm University, SE-10691 Stockholm, Sweden
[8]SINTEF Energy Research, 7465 Trondheim, Norway
[9]Department of Energy and Process Engineering, NTNU, 7491 Trondheim, Norway
**Correspondence:** Xiang-Yu Li (xiang.yu.li@su.se), December 13, 2018, Revision: 1.200

**Abstract.**

Condensational growth of cloud droplets due to supersaturation fluctuations is investigated by solving the hydrodynamic and thermodynamic equations using direct numerical simulations with droplets being modeled as Lagrangian particles. The supersaturation field is calculated directly by simulating the temperature and water vapor fields instead of being treated as a passive scalar. Thermodynamic feedbacks to the fields due to condensation are also included for completeness. We find that the width of droplet size distributions increases with time, which is contrary to the classical theory without supersaturation fluctuations, where condensational growth leads to progressively narrower size distributions. Nevertheless, in agreement with earlier Lagrangian stochastic models of the condensational growth, the standard deviation of the surface area of droplets increases as $t^{1/2}$. Also, for the first time, we explicitly demonstrate that the time evolution of the size distribution is sensitive to the Reynolds number, but insensitive to the mean energy dissipation rate. This is shown to be due to the fact that temperature fluctuations and water vapor mixing ratio fluctuations increases with increasing Reynolds number, therefore the resulting supersaturation fluctuations are enhanced with increasing Reynolds number. Our simulations may explain the broadening of the size distribution in stratiform clouds qualitatively, where the mean updraft velocity is almost zero.

## 1 Introduction

The growth of cloud droplets is dominated by two processes: condensation and collection. Condensation of water vapor on active cloud condensation nuclei is important in the size range from the activation size of aerosol particles to about a radius of $10\,\mu\mathrm{m}$ (Pruppacher and Klett, 2012; Lamb and Verlinde, 2011). Since the rate of droplet growth by condensation is inversely proportional to the droplet radius, large droplets grow slower than smaller ones. This generates narrower size distributions (Lamb and Verlinde, 2011). To form rain droplets in warm clouds, small droplets must grow to about $50\,\mu\mathrm{m}$ in radius within

15–20 minutes (Pruppacher and Klett, 2012; Devenish et al., 2012; Grabowski and Wang, 2013; Seinfeld and Pandis, 2016). Therefore, collection, a widely accepted microscopical mechanism, has been proposed to explain the rapid formation of rain droplets (Saffman and Turner, 1956; Berry and Reinhardt, 1974; Shaw, 2003; Grabowski and Wang, 2013). However, collection can only become active when the size distribution reaches a certain width.

Hudson and Svensson (1995) observed a broadening of the droplet size distribution in Californian marine stratus, which was contrary to the classical theory of condensational growth (Yau and Rogers, 1996). The increasing width of droplet size distributions were further observed by Pawlowska et al. (2006) and Siebert and Shaw (2017b). The contradiction between the observed broadening width and the theoretical narrowing width in the absence of turbulence has stimulated several studies. The classical treatment of diffusion-limited growth assumes that supersaturation depends only on average temperature and water

mixing ratio. Since fluctuations of temperature and the water mixing ratio are affected by turbulence, the supersaturation fluctuations are inevitably subjected to turbulence. Naturally, condensational growth due to supersaturation fluctuations became the focus (Sedunov, 1965; Kabanov and Mazin, 1970; Cooper, 1989; Srivastava, 1989; Korolev, 1995; Khvorostyanov and Curry, 1999; Sardina et al., 2015; Grabowski and Abade, 2017). The supersaturation fluctuations are particularly important for understanding the condensational growth of cloud droplets in stratiform clouds, where the updraft velocity of the parcel is almost

zero (Hudson and Svensson, 1995; Korolev, 1995). When the mean updraft velocity is not zero, there could be a competition between mean updraft velocity and supersaturation fluctuations. This may diminish the role of supersaturation fluctuations (Sardina et al., 2018).

Condensational growth due to supersaturation fluctuations was first recognized by Srivastava (1989), who criticized the use of a volume-averaged supersaturation and proposed a randomly distributed supersaturation field. Cooper (1989) proposed

that droplets moving in clouds are exposed to a varying supersaturation field. This results in broadening of droplets size distribution due to supersaturation fluctuations. Grabowski and Wang (2013) called the mechanism of Cooper (1989) the eddy-hopping mechanism, which was then investigated by Grabowski and Abade (2017). Using direct numerical simulations (DNS), Vaillancourt et al. (2002) found that the mean energy dissipation rate of turbulence has a negligible effect on condensational growth and attributed this to the decorrelation between the supersaturation and the droplet size. Paoli and Shariff (2009) consid-

ered three-dimensional (3-D) turbulence as well as stochastically forced temperature and vapor fields with a focus on statistical modeling for large-eddy simulations. They found that supersaturation fluctuations due to turbulence mixing are responsible for the broadening of the droplet size distribution. Lanotte et al. (2009) conducted 3-D DNS for condensational growth by only solving a passive scalar equation for the supersaturation and concluded that the width of the size distribution increases with increasing Reynolds number. Sardina et al. (2015) extended the DNS of Lanotte et al. (2009) to higher Reynolds number and

found that the variance of the size distribution increases in time. In a similar manner as Sardina et al. (2015), Siewert et al. (2017) modelled the supersaturation field as a passive scalar coupled to the Lagrangian particles and found that their results can be reconciled with those of earlier numerical studies by noting that the droplet size distribution broadens with increasing Reynolds number (Paoli and Shariff, 2009; Lanotte et al., 2009; Sardina et al., 2015). Neither Sardina et al. (2015) nor Siewert et al. (2017) solved the thermodynamics that determine the supersaturation field. Both Saito and Gotoh (2017) and

Chen et al. (2018) solved the thermodynamics equations governing the supersaturation field. However, since collection was

also included in their work, one cannot clearly identify the roles of turbulence on collection or condensational growth, nor can one compare their results with Lagrangian stochastic models (Sardina et al., 2015; Siewert et al., 2017) related to condensational growth.

Recent laboratory experiments and observations about cloud microphysics also confirm the notion that supersaturation fluctuations may play an important role in broadening the size distribution of cloud droplets. The laboratory studies of Chandrakar et al. (2016) and Desai et al. (2018) suggested that supersaturation fluctuations in the low aerosol number concentration limit are likely of leading importance for the onset of precipitation. The condensational growth due to supersaturation fluctuations seems to be more sensitive to the integral scale of turbulence (Götzfried et al., 2017). Siebert and Shaw (2017a) measured the variability of temperature, water vapor mixing ratio, and supersaturation in warm clouds and support the notion that both aerosol particle activation and droplet growth take place in the presence of a broad distribution of supersaturation (Hudson and Svensson, 1995; Brenguier et al., 1998; Miles et al., 2000; Pawlowska et al., 2006). The challenge is now how to interpret the observed broadening of droplet size distribution in warm clouds. How does turbulence drive fluctuations of the scalar fields (temperature and water vapor mixing ratio) and therefore affect the broadening of droplet size distributions (Siebert and Shaw, 2017a)?

In an attempt to answer this question, we conduct 3-D DNS experiments of condensational growth of cloud droplets, where turbulence, thermodynamics, feedback from droplets to the fields via the condensation rate and buoyancy force are all included. The main aim is to investigate how supersaturation fluctuations affect the droplet size distribution. We particularly focus on the time evolution of the size distribution $f(r,t)$ and its dependency on small and large scales of turbulence. We then compare our simulation results with Lagrangian stochastic models (Sardina et al., 2015; Siewert et al., 2017). For the first time, the stochastic model and simulation results from the complete set of equations governing the supersaturation field are compared.

## 2   Numerical model

We now discuss the basic equations where we combine the Eulerian description of the density ($\rho$), turbulent velocity ($\boldsymbol{u}$), temperature ($T$), and water vapor mixing ratio ($q_v$) with the Lagrangian description of the ensemble of cloud droplets. The water vapor mixing ratio $q_v$ is defined as the ratio between the mass density of water vapor and dry air. Droplets are treated as superparticles. A superparticle represents an ensemble of droplets, whose mass, radius, and velocity are the same as those of each individual droplet within it (Shima et al., 2009; Johansen et al., 2012; Li et al., 2017). For condensational growth, the superparticle approach (Li et al., 2017) is the same as the Lagrangian point-particle approach (Kumar et al., 2014) since there is no interactions among droplets. Nevertheless, we still use the superparticle approach so that we can include more processes like collection (Li et al., 2017, 2018) in future. Another reason to adopt superparticle approach is that it can be easily adapted to conduct Large-eddy simulations with appropriate sub-grid scale models (Grabowski and Abade, 2017). To investigate the condensational growth of cloud droplets that experience fluctuating supersaturation, we track each individual superparticle in a Lagrangian manner. The motion of each superparticle is governed by the momentum equation for inertial particles. The supersaturation field in the simulation domain is determined by $T(\boldsymbol{x},t)$ and $q_v(\boldsymbol{x},t)$ transported by turbulence.

Lagrangian droplets are exposed in different supersaturation fields. Therefore, droplets either grow by condensation or shrink by evaporation depending on the local supersaturation field. This phase transition generates a buoyancy force, which in turn affects the turbulent kinetic energy, $T(\boldsymbol{x},t)$, and $q_v(\boldsymbol{x},t)$. PENCIL CODE is used to conduct all the simulations.

## 2.1 Equations of motion for Eulerian fields

The background air flow is almost incompressible and thus obeys the Boussinesq approximation. Its density $\rho(\boldsymbol{x},t)$ is governed by the continuity equation and velocity $\boldsymbol{u}(\boldsymbol{x},t)$ by Navier-Stokes equation. The temperature $T(\boldsymbol{x},t)$ of the background air flow is determined by the energy equation with a source term due to the latent heat release. The water vapor mixing ratio $q_v(\boldsymbol{x},t)$ is transported by the background air flow. The Eulerian equations are given by

$$\frac{\partial \rho}{\partial t} + \boldsymbol{\nabla} \cdot (\rho \boldsymbol{u}) = S_\rho, \tag{1}$$

$$\frac{D\boldsymbol{u}}{Dt} = \boldsymbol{f} - \rho^{-1}\boldsymbol{\nabla}p + \rho^{-1}\boldsymbol{\nabla} \cdot (2\nu\rho\mathsf{S}) + B\boldsymbol{e}_z + \boldsymbol{S}_u, \tag{2}$$

$$\frac{DT}{Dt} = \kappa\nabla^2 T + \frac{L}{c_p}C_d, \tag{3}$$

$$\frac{Dq_v}{Dt} = D\nabla^2 q_v - C_d, \tag{4}$$

where $D/Dt = \partial/\partial t + \boldsymbol{u}\cdot\boldsymbol{\nabla}$ is the material derivative, $\boldsymbol{f}$ is a random forcing function (Haugen et al., 2004), $\nu$ is the kinematic viscosity of air, $\mathsf{S}_{ij} = \frac{1}{2}(\partial_j u_i + \partial_i u_j) - \frac{1}{3}\delta_{ij}(\partial_k u_k)$ is the traceless rate-of-strain tensor, $p$ is the gas pressure, $\rho$ is the gas density, $c_p$ is the specific heat at constant pressure, $L$ is the latent heat, $\kappa$ is the thermal diffusivity of air, $C_d$ is the condensation rate, $B$ is the buoyancy, $\boldsymbol{e}_z$ is the unit vector in the $z$ direction (vertical direction), and $D$ is the diffusivity of water vapor.

To avoid global transpose operations associated with calculating Fourier transforms for solving the nonlocal equation for the pressure in strictly incompressible calculations, we solve here instead the compressible Navier-Stokes equations using high-order finite differences. The sound speed $c_s$ obeys $c_s^2 = \gamma p/\rho$, where $\gamma = c_p/c_v = 7/5$ is the ratio between specific heats, $c_p$ and $c_v$, at constant pressure and constant volume, respectively. We set the sound speed as $5\,\mathrm{m\,s^{-1}}$ to simulate the nearly incompressible atmospheric air flow, resulting in a Mach number of 0.06 when $u_{\mathrm{rms}} = 0.27\,\mathrm{m\,s^{-1}}$, where $u_{\mathrm{rms}}$ is the rms

velocity. Such a configuration, with so small Mach number, is almost equivalent to an incompressible flow. It is worth noting that the temperature determining the compressibility of the flow is constant and independent of the temperature field of the gas flow governed by Equation (3). Also, since the gas flow is almost incompressible and its mass density is much smaller than the one of the droplet, there is no mass exchange between the gas flow and the droplet, i.e., the density of the gas flow $\rho(\boldsymbol{x},t)$ is not affected by $T(\boldsymbol{x},t)$. Thus, the source terms $S_\rho$ and $\boldsymbol{S}_u$ in Equations (1) and (2) are neglected (Krüger et al., 2017).

The buoyancy $B(\boldsymbol{x},t)$ depends on the temperature $T(\boldsymbol{x},t)$, water vapor mixing ratio $q_v(\boldsymbol{x},t)$, and the liquid mixing ratio $q_l$ (Kumar et al., 2014),

$$B(\boldsymbol{x},t) = g(T'/T + \alpha q_v' - q_l), \tag{5}$$

where $\alpha = M_a/M_v - 1 \approx 0.608$ when $M_a$ and $M_v$ are the molar masses of air and water vapor, respectively. The amplitude of the gravitational acceleration is given by $g$. The liquid water mixing ratio is the ratio between the mass density of liquid water and the dry air and is defined as

$$q_l(\boldsymbol{x},t) = \frac{4\pi\rho_l}{3\rho_a(\Delta x)^3} \sum_{j=1}^{N_\triangle} r(t)^3 = \frac{4\pi\rho_l}{3\rho_a} \sum_{j=1}^{N_\triangle} f(r,t)r(t)^3 \delta r, \tag{6}$$

where $\rho_l$ and $\rho_a$ are the liquid water density and the reference mass density of dry air. $N_\triangle$ is the total number of droplets in a cubic grid cell with volume $(\Delta x)^3$, where $\Delta x$ is the one-dimensional size of the grid box. The temperature fluctuations are given by

$$T'(\boldsymbol{x},t) = T(\boldsymbol{x},t) - T_{\text{env}}, \tag{7}$$

and the water vapor mixing ratio fluctuations by

$$q_v'(\boldsymbol{x},t) = q_v(\boldsymbol{x},t) - q_{v,\text{env}}. \tag{8}$$

We adopt the same method as in Kumar et al. (2014), where the mean environmental temperature $T_{\text{env}}$ and water vapor mixing ratio $q_{v,\text{env}}$ do not change in time. This assumption is plausible in the circumstance that we do not consider the entrainment, i.e., there is only mass and energy transfer between liquid water and water vapor. The condensation rate $C_d$ (Vaillancourt et al., 2001) is given by

$$C_d(\boldsymbol{x},t) = \frac{4\pi\rho_l G}{\rho_a(\Delta x)^3} \sum_{j=1}^{N_\triangle} s(\boldsymbol{x},t) r(t) = \frac{4\pi\rho_l G}{\rho_a} \sum_{j=1}^{N_\triangle} s(\boldsymbol{x},t) f(\boldsymbol{x},t) r(t) \delta r, \tag{9}$$

where $G$ is the condensation parameter (in units of $\text{m}^2\,\text{s}^{-1}$), which depends weakly on temperature and pressure and is here assumed to be constant (Lamb and Verlinde, 2011). The supersaturation $s$ is defined as the ratio between the vapor pressure $e_v$ and the saturation vapor pressure $e_s$,

$$s = \frac{e_v}{e_s} - 1. \tag{10}$$

Using the ideal gas law, Equation (10) can be expressed as,

$$s = \frac{\rho_v R_v T}{\rho_{vs} R_v T} - 1 = \frac{\rho_v}{\rho_{vs}} - 1. \tag{11}$$

In terms of the water vapor mixing ratio $q_v = \rho_v/\rho_a$ and saturation water vapor mixing ratio $q_{vs} = \rho_{vs}/\rho_a$, Equation (11) can be written as:

$$s(\boldsymbol{x}) = \frac{q_v(\boldsymbol{x},t)}{q_{vs}(T)} - 1. \tag{12}$$

Here $\rho_v$ is the mass density of water vapor and $\rho_{vs}$ the mass density of saturated water vapor, and $q_{vs}(T)$ is the saturation water vapor mixing ratio at temperature $T$ and can be determined by the ideal gas law,

$$q_{vs}(T) = \frac{e_s(T)}{R_v \rho_a T}. \tag{13}$$

The saturation vapor pressure $e_s$ over liquid water is the partial pressure due to the water vapor when an equilibrium state of evaporation and condensation is reached for a given temperature. It can be determined by the Clausius-Clapeyron equation, which determines the change of $e_s$ with temperature $T$. Assuming constant latent heat $L$, $e_s$ is approximated as (Yau and Rogers, 1996; Götzfried et al., 2017)

$$e_s(T) = c_1 \exp(-c_2/T), \tag{14}$$

where $c_1$ and $c_2$ are constants adopted from page 14 of Yau and Rogers (1996). We refer to Table 1 for all the thermodynamics constants. In the present study, the updraft cooling is omitted. Therefore, the assumption of constant latent heat $L$ is plausible.

## 2.2 Lagrangian model for cloud droplets

In addition to the Eulerian fields described in Section 2.1 we treat cloud droplets as Lagrangian particles. In the PENCIL CODE, they are invoked as non-interacting superparticles.

### 2.2.1 Kinetics of cloud droplets

Each superparticle is treated as a Lagrangian point-particle, where one solves for the particle position $\boldsymbol{x}_i$,

$$\frac{d\boldsymbol{x}_i}{dt} = \boldsymbol{V}_i, \tag{15}$$

and its velocity $\boldsymbol{V}_i$ via

$$\frac{d\boldsymbol{V}_i}{dt} = \frac{1}{\tau_i}(\boldsymbol{u} - \boldsymbol{V}_i) + g\boldsymbol{e}_z, \tag{16}$$

in the usual way; see (Li et al., 2017) for details. Here, $\boldsymbol{u}$ is the fluid velocity at the position of the superparticle, $\tau_i$ is the particle inertial response or stopping time of a droplet $i$ and is given by

$$\tau_i = 2\rho_l r_i^2/[9\rho\nu\, D(\mathrm{Re}_i)]. \tag{17}$$

The correction factor (Schiller and Naumann, 1933; Marchioli et al., 2008),

$$D(\mathrm{Re}_i) = 1 + 0.15\,\mathrm{Re}_i^{2/3}, \tag{18}$$

models the effect of non-zero particle Reynolds number $\mathrm{Re}_i = 2r_i|\boldsymbol{u} - \boldsymbol{V}_i|/\nu$. This is a widely used approximation, although it does not correctly reproduce the small-$\mathrm{Re}_i$ correction to Stokes formula (Veysey and Goldenfeld, 2007).

### 2.2.2 Condensational growth of cloud droplets

The condensational growth of the particle radius $r_i$ is governed by (Pruppacher and Klett, 2012; Lamb and Verlinde, 2011)

$$\frac{\mathrm{d}r_i}{\mathrm{d}t} = \frac{Gs\,(\boldsymbol{x}_i,t)}{r_i}. \tag{19}$$

## 3 Experimental setup

### 3.1 Initial configurations

The initial values of the water vapor mixing ratio $q_v(\boldsymbol{x}, t=0) = 0.0157\,\mathrm{kg \cdot kg^{-1}}$ and temperature $T(\boldsymbol{x}, t=0) = 292\,\mathrm{K}$ are matched to the ones obtained in the CARRIBA experiments (Katzwinkel et al., 2014), which are the same as those in Götzfried et al. (2017). With this configuration, we obtain $s(\boldsymbol{x}, t=0) = 2\%$, which means that the water vapor is initially supersaturated. The time step of the simulations presented here is governed by the smallest time scale in the present configuration, which is the particle stopping time defined in Equation (17). The thermodynamic time scale is much larger than the turbulent one. Table 1 shows the list of thermodynamic parameters used in the present study.

Initially, 10 $\mu$m-sized droplets with zero velocity are randomly distributed in the simulation domain. The mean number density of droplets, which is constant in time since droplet collections are not considered, is $n_0 = 2.5 \times 10^8\,\mathrm{m^{-3}}$. This gives an initial liquid water content, $\int_0^\infty f(r, t=0)\, r^3\, \mathrm{d}r$, which is $0.001\,\mathrm{kg\,m^{-3}}$. The simulation domain is a cube of size $L_x = L_y = L_z$, the values of which are given in Table 2. The number of superparticles $N_s$ satisfies $N_s/N_{\mathrm{grid}} \approx 0.1$, where $N_{\mathrm{grid}}$ is the number of lattices depending on the spatial resolution of the simulations. Setting $N_s/N_{\mathrm{grid}} \approx 0.1$, on one hand, is still within the convergence range $N_s/N_{\mathrm{grid}} \approx 0.05$ (Li et al., 2018). On the other hand, it can mimic the diluteness of the atmospheric cloud system, where there are about 0.1 droplets per cubic Kolmogorov scale. This configuration results in $N_{s,128} = 244140$ when $N_{\mathrm{grid}} = 128^3$.

### 3.2 DNS

We conduct high resolution simulations for different Taylor micro-scale Reynolds number $\mathrm{Re}_\lambda$ and mean energy dissipation rate $\bar{\epsilon}$ (see Table 2 for details of the simulations). The Taylor micro-scale Reynolds number is defined as $\mathrm{Re}_\lambda \equiv u_{\mathrm{rms}}^2 \sqrt{5/(3\nu\bar{\epsilon})}$. For simulations with different values of $\bar{\epsilon}$ at fixed $\mathrm{Re}_\lambda$, we vary both the domain size $L_x$ ($L_y = L_z = L_x$) and the amplitude of the forcing $f_0$. As for fixed $\bar{\epsilon}$, $\mathrm{Re}_\lambda$ is varied by solely changing the domain size, which in turn changes $u_{\mathrm{rms}}$. In all simulations, we use for the Prandtl number $\mathrm{Pr} = \nu/\kappa = 1$ and for the Schmidt number $Sc = \nu/D = 0.6$. For our simulations with $N_{\mathrm{grid}} = 512^3$ meshpoints, the code computes 55,000 time steps in 24 hours wall-clock time using 4096 cores. For $N_{\mathrm{grid}} = 128^3$ meshpoints, the code computes 4.5 million time steps in 24 hours wall-clock time using 512 cores.

## 4 Results

Figure 1(a) shows time-averaged turbulent kinetic-energy spectra for different values of $\bar{\epsilon}$ at fixed $\mathrm{Re}_\lambda \approx 130$. Since the abscissa in the figures is normalized by $k_\eta = 2\pi/\eta$, the different spectra shown in Figure 1(a) collapse onto a single curve. Here, $\eta$ is the Kolmogorov length scale. Figure 1(b) shows the time-averaged turbulent kinetic-energy spectra for different values of $\mathrm{Re}_\lambda$ at fixed $\bar{\epsilon} \approx 0.039\,\mathrm{m^2 s^{-3}}$. For larger Reynolds numbers the spectra extend to smaller wavenumbers. A flat profile corresponds to Kolmogorov scaling (Pope, 2000) when the energy spectrum is compensated by $\bar{\epsilon}^{-2/3} k^{5/3}$. For the largest $\mathrm{Re}_\lambda$ in our simulations ($\mathrm{Re}_\lambda = 130$), the inertial range extends for about a decade in $k$-space.

**Table 1.** List of constants for the thermodynamics: see text for explanations of symbols.

| Quantity | Value |
|---|---|
| $\nu\ (\mathrm{m^2\,s^{-1}})$ | $1.5 \times 10^{-5}$ |
| $\kappa\ (\mathrm{m^2\,s^{-1}})$ | $1.5 \times 10^{-5}$ |
| $D\ (\mathrm{m^2\,s^{-1}})$ | $2.55 \times 10^{-5}$ |
| $G\ (\mathrm{m^2\,s^{-1}})$ | $1.17 \times 10^{-10}$ |
| $c_1\ (\mathrm{Pa})$ | $2.53 \times 10^{11}$ |
| $c_2\ (\mathrm{K})$ | 5420 |
| $L\ (\mathrm{J \cdot kg^{-1}})$ | $2.5 \times 10^6$ |
| $c_\mathrm{p}\ (\mathrm{J \cdot kg^{-1}K^{-1}})$ | 1005.0 |
| $R_v\ (\mathrm{J \cdot kg^{-1}K^{-1}})$ | 461.5 |
| $M_a\ (\mathrm{g \cdot mol^{-1}})$ | 28.97 |
| $M_v\ (\mathrm{g \cdot mol^{-1}})$ | 18.02 |
| $\rho_a\ (\mathrm{kg \cdot m^{-3}})$ | 1 |
| $\rho_l\ (\mathrm{kg \cdot m^{-3}})$ | 1000 |
| $\alpha$ | 0.608 |
| $\mathrm{Pr} = \nu/\kappa$ | 1 |
| $Sc = \nu/D$ | 0.6 |
| $q_v(\boldsymbol{x}, t=0)\ (\mathrm{kg \cdot kg^{-1}})$ | 0.0157 |
| $q_{v,\mathrm{env}}\ (\mathrm{kg \cdot kg^{-1}})$ | 0.01 |
| $T(\boldsymbol{x}, t=0)\ (\mathrm{K})$ | 292 |
| $T_\mathrm{env}\ (\mathrm{K})$ | 293 |

**Table 2.** Summary of the simulations; see text for explanation of symbols.

| Run | $f_0$ | $L_x\ (\mathrm{m})$ | $N_\mathrm{grid}$ | $N_\mathrm{s}$ | $u_\mathrm{rms}\ (\mathrm{m\,s^{-1}})$ | $\mathrm{Re}_\lambda$ | $\bar{\epsilon}\ (\mathrm{m^2 s^{-3}})$ | $\eta \cdot 10^{-4}\ (\mathrm{m})$ | $\tau_\eta\ (\mathrm{s})$ | $\tau_\mathrm{L}\ (\mathrm{s})$ | $\tau_\mathrm{s}\ (\mathrm{s})$ | Da |
|---|---|---|---|---|---|---|---|---|---|---|---|---|
| A | 0.02 | 0.125 | $128^3$ | $N_\mathrm{s,128}$ | 0.16 | 45 | 0.039 | 5.4 | 0.020 | 0.25 | 0.014 | 0.053 |
| B | 0.02 | 0.25 | $256^3$ | $2^3 N_\mathrm{s,128}$ | 0.22 | 78 | 0.039 | 5.4 | 0.020 | 0.37 | 0.014 | 0.081 |
| C | 0.02 | 0.5 | $512^3$ | $2^6 N_\mathrm{s,128}$ | 0.28 | 130 | 0.039 | 5.4 | 0.020 | 0.58 | 0.014 | 0.125 |
| D | 0.014 | 0.6 | $512^3$ | $2^6 N_\mathrm{s,128}$ | 0.24 | 135 | 0.019 | 6.5 | 0.028 | 0.81 | 0.014 | 0.174 |
| E | 0.007 | 0.8 | $512^3$ | $2^6 N_\mathrm{s,128}$ | 0.17 | 138 | 0.005 | 8.9 | 0.053 | 1.47 | 0.014 | 0.312 |

Next we inspect the response of thermodynamics to turbulence. In Figure 2, we show time series of fluctuations of temperature $T_\mathrm{rms}$, water vapor mixing ratio $q_{v,\mathrm{rms}}$, buoyancy force $B_\mathrm{rms}$, and the supersaturation $s_\mathrm{rms}$. All quantities reach a statistically steady state within a few seconds. The steady state values of $T_\mathrm{rms}$, $q_{v,\mathrm{rms}}$, and $s_\mathrm{rms}$ increase with increasing $\mathrm{Re}_\lambda$ approximately linearly, and vary hardly at all with $\bar{\epsilon}$. On the other hand. $B_\mathrm{rms}$ changes only by a few percent as $\mathrm{Re}_\lambda$ or $\bar{\epsilon}$ vary.

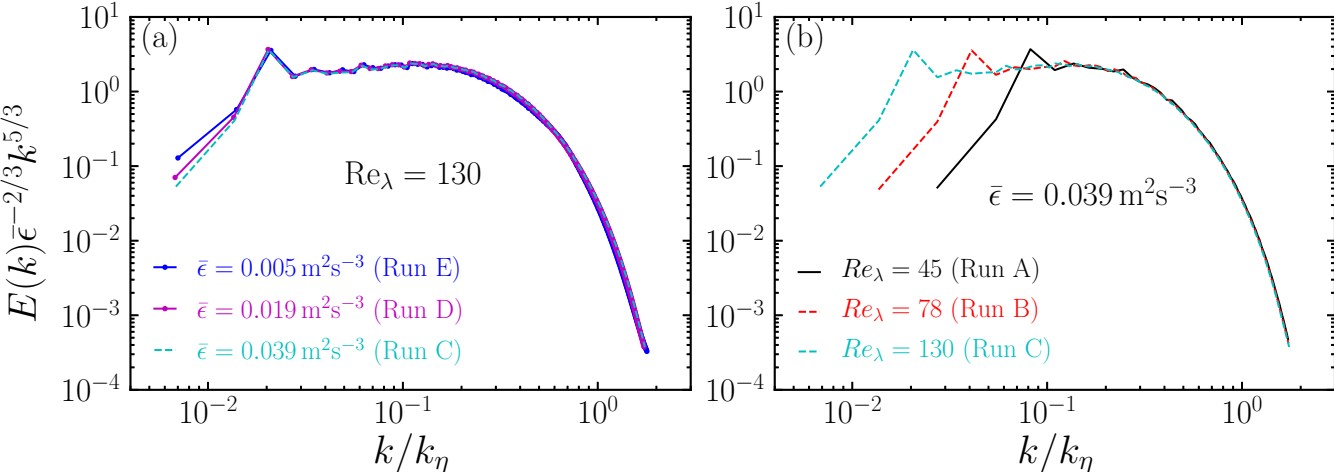

**Figure 1.** Time-averaged kinetic energy spectra of the turbulence gas flow for (a) different $\bar{\epsilon} = 0.005\,\mathrm{m^2s^{-3}}$ (blue dash-dotted line), 0.019 (magenta dash-dotted line) and 0.039 (black solid line) at fixed $\mathrm{Re}_\lambda = 130$ (see Runs C, D, and E in Table 2 for details) and for (b) different $\mathrm{Re}_\lambda = 45$ (black solid line), 78 (red dashed line), and 130 (cyan dashed line) at fixed $\bar{\epsilon} = 0.039\mathrm{m^2s^{-3}}$ (see Runs A, B, and C in Table 2 for details).

Note, however, that the buoyancy force is only about 0.3% of the fluid acceleration. This is because $T_{\mathrm{rms}}$ is small (about $0.1\,\mathrm{K}$ in the present study). Therefore, the effect of the buoyancy force should indeed be small.

When changing $\bar{\epsilon}$ while keeping $\mathrm{Re}_\lambda$ fixed, the Kolmogorov scales of turbulence varies. Therefore, the various fluctuations quoted above are insensitive to the small scales of turbulence. However, when varying $\mathrm{Re}_\lambda$ while keeping $\bar{\epsilon}$ fixed, their rms

values change, which is due to large scales of turbulence. Indeed, temperature fluctuations are driven by the large scales of turbulence, which affects the supersaturated vapor pressure $q_{vs}$ via the Clausius-Clapeyron equation; see Equation (13). Therefore, supersaturation fluctuations result from both temperature fluctuations and water vapor fluctuations via Equation (12). Both $q_{v,\mathrm{rms}}$ and $T_{\mathrm{rms}}$ increase with increasing $\mathrm{Re}_\lambda$, resulting in larger fluctuations of $s$. Supersaturation fluctuations, in turn, affect $T$ and $q_v$ via the condensation rate $C_d$.

Our goal is to investigate the condensational growth of cloud droplets due to supersaturation fluctuations. Figure 3 shows the time evolution of droplet size distributions for different configurations. The conventional understanding is that condensational growth leads to a narrow size distribution (Pruppacher and Klett, 2012; Lamb and Verlinde, 2011). However, supersaturation fluctuations broaden the distribution. More importantly, the width of the size distribution increases with increasing $\mathrm{Re}_\lambda$, but *decreases* slightly with increasing $\bar{\epsilon}$ over the range studied here. This is consistent with the results shown in Figure 2 in that

supersaturation fluctuations are sensitive to $\mathrm{Re}_\lambda$ but are insensitive to $\bar{\epsilon}$. In atmospheric clouds, $\mathrm{Re}_\lambda \approx 10^4$, which may result in an even broader size distribution.

We further quantify the variance of the size distribution by investigating the time evolution of the standard deviation of the droplet surface area $\sigma_A$ for different configurations. In terms of the droplet surface area $A_i$ ($A_i \propto r_i^2$), Equation (19) can be

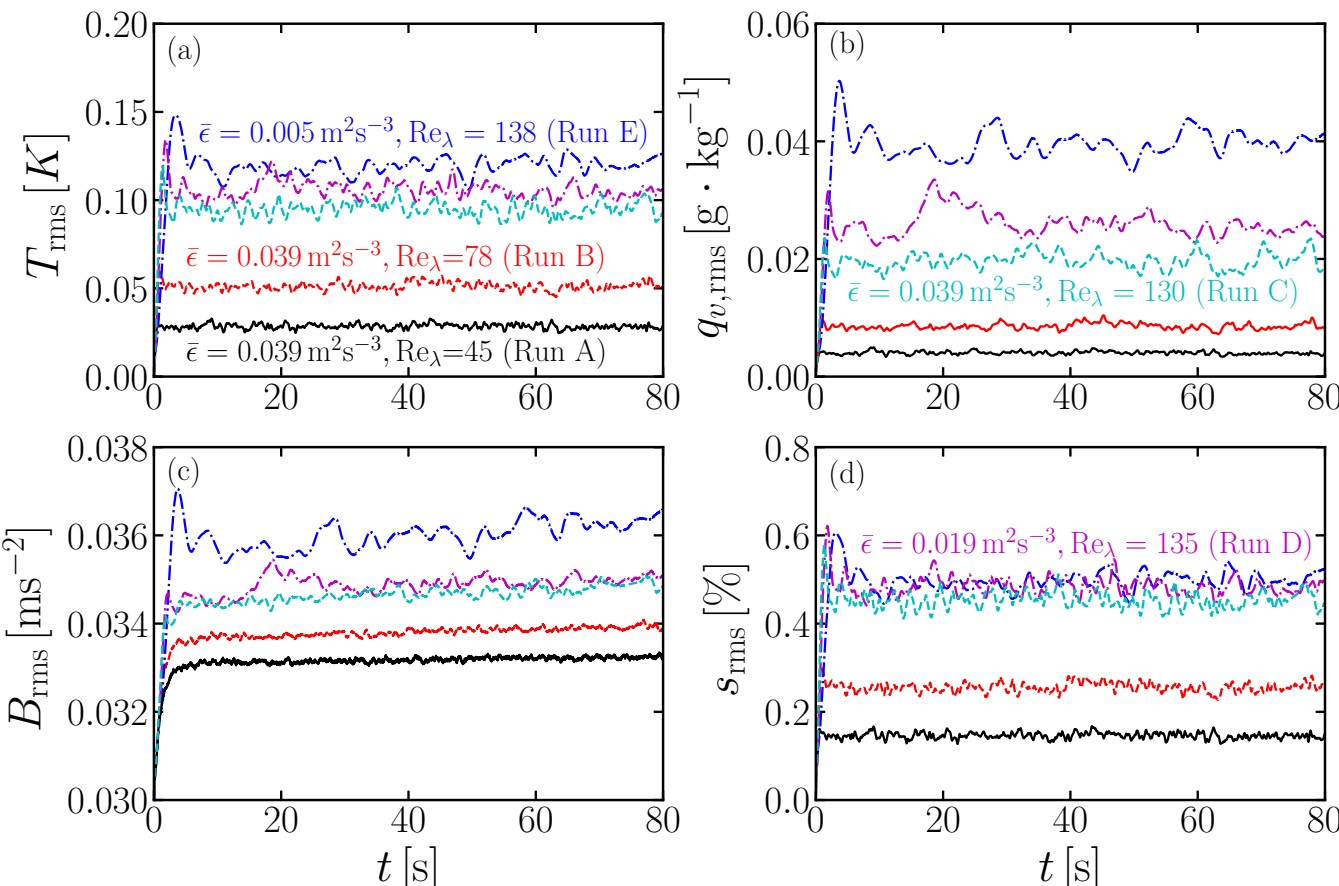

**Figure 2.** Time series of the field quantities: (a) $T_{\mathrm{rms}}$, (b) $q_v,\mathrm{rms}$, (c) $B_{\mathrm{rms}}$, and (d) $s_{\mathrm{rms}}$. Same simulations as in Figure 1.

written as

$$\frac{dA_i}{dt} = 2Gs. \tag{20}$$

It can be seen from Equation (20) that the evolution of the surface area is analogous to Brownian motion, indicating that its standard deviation $\sigma_A \propto \sqrt{t}$. A more detailed stochastic model for $\sigma_A$ is developed by Sardina et al. (2015). Based on Equation (19), $\sigma_A$ is given by

$$\frac{d\sigma_A^2}{dt} = \frac{d}{dt}\left\langle A'^2 \right\rangle = \frac{d}{dt}\left\langle A^2 - \langle A \rangle^2 \right\rangle = 4G\left\langle s'A' \right\rangle. \tag{21}$$

Sardina et al. (2015) adopted a Langevin equation to model the supersaturation field and the vertical velocity of droplets, resulting in the scaling law:

$$\sigma_A \sim C(\tau_L, \tau_s, \mathrm{Re}_\lambda) t^{1/2}, \tag{22}$$

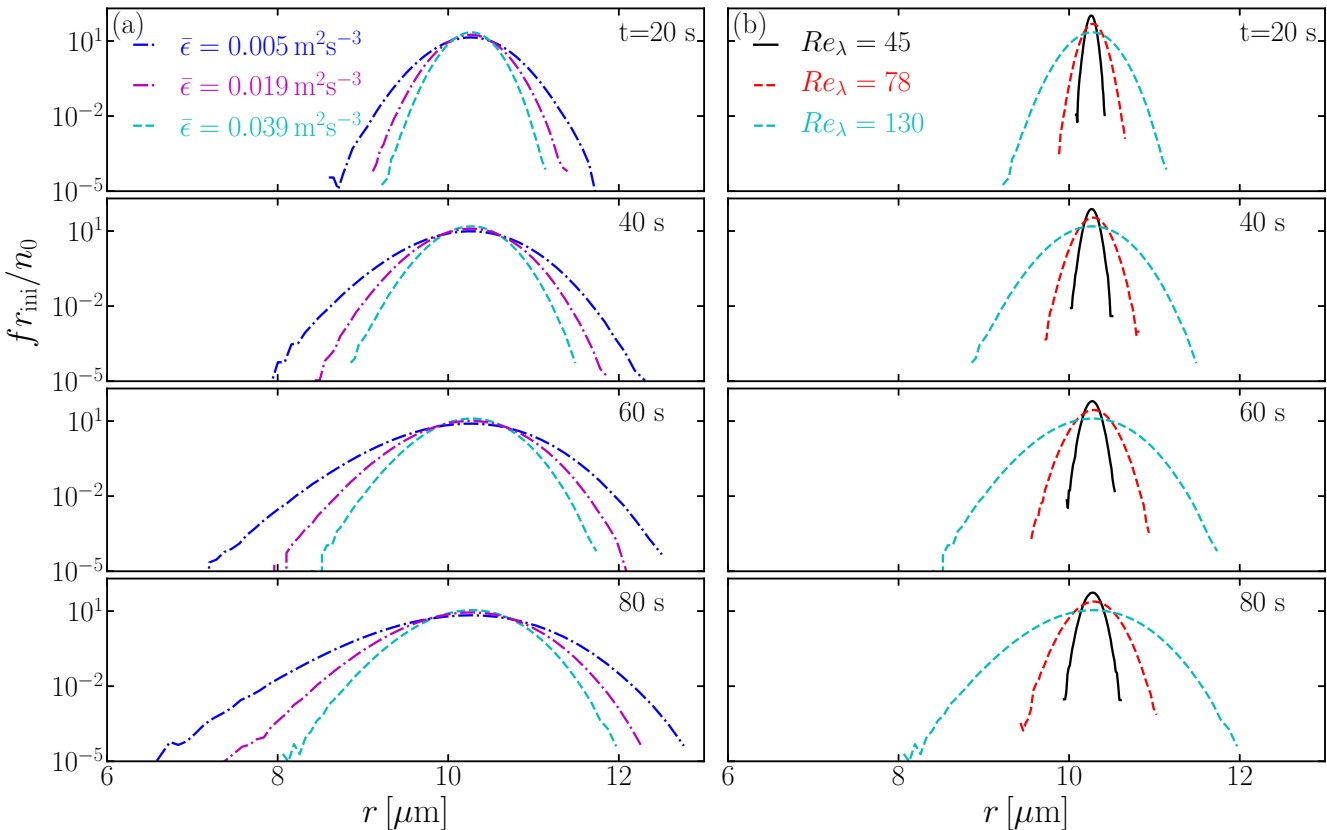

**Figure 3.** Comparison of the time evolution of droplet size distributions for different (a) $\bar{\epsilon}$ at $\mathrm{Re}_\lambda = 130$ (Runs C, D, and E in Table 2) and (b) $\mathrm{Re}_\lambda$ at $\bar{\epsilon} = 0.039\,\mathrm{m^2s^{-3}}$ (Runs A, B, and C in Table 2). Same simulations as in Figure 1.

where $C(\tau_L, \tau_{\mathrm{s,Re}_\lambda})$ is a constant for given $\tau_L$, $\tau_{\mathrm{s}}$, and $\mathrm{Re}_\lambda$. Under the assumptions that $\tau_{\mathrm{s}} \ll T_L$ and a negligible influence on the macroscopic observables from small-scale turbulent motions, Sardina et al. (2015) obtained an analytical expression for $\sigma_A$ as:

$$\sigma_A \sim \tau_{\mathrm{s}} \mathrm{Re}_\lambda t^{1/2}, \tag{23}$$

5   where $\tau_{\mathrm{s}}$ is the phase transition time scale given by

$$\tau_{\mathrm{s}}^{-1}(t) = 4\pi G \int_0^\infty r f\, dr, \tag{24}$$

and $\tau_L$ is the turbulence integral time scale. The model proposed that condensational growth of cloud droplets depends only on $\mathrm{Re}_\lambda$ and is independent of $\bar{\epsilon}$. In terms of the size distribution $f(r,t)$, $\sigma_A$ can be given as:

$$\sigma_A = \sqrt{a_4 - a_2^2}, \tag{25}$$

where $a_\zeta$ is the moment of the size distribution, which is defined as:

$$a_\zeta = \int\limits_0^\infty f \, r^\zeta \, \mathrm{d}r \Big/ \int\limits_0^\infty f \, \mathrm{d}r. \tag{26}$$

Here, $\zeta$ is a positive integer. As shown in Figure 4, the time evolution of $\sigma_A$ agrees with the prediction $\sigma_A \propto t^{1/2}$. Sardina et al. (2015) and Siewert et al. (2017) solved the passive scalar equation of $s$ without considering fluctuations of $T$ and $q_v$. Feedbacks

to flow fields from cloud droplets were also neglected. They found good agreement between the DNS and the stochastic model. Comparing with Sardina et al. (2015) and Siewert et al. (2017), our study solve the complete sets of the thermodynamics of supersaturation. It is remarkable that a good agreement between the stochastic model and our DNS is observed. This indicates that the stochastic model is robust. On the other hand, modeling supersaturation fluctuations using the passive scalar equation seems to be sufficient for the Reynolds numbers considered in this study. We recall that $\tau_s$ in Equation (23) is constant. In

the present study, $\tau_s$ is determined by Equation (24). Therefore, $\tau_s$ varies with time as shown in the inset of Figure 4(a). Nevertheless, since the variation of $\tau_s$ is small, we still observe $\sigma_A \sim t^{1/2}$ except for the initial phase of the evolution, where $s(t = 0) = 2\%$.

Comparing panels (a) and (b) of Figure 4, it is clear that changing $\mathrm{Re}_\lambda$ has a much larger effect on $\sigma_A$ than changing $\bar{\epsilon}$. In fact, as $\bar{\epsilon}$ is increased by a factor of about 8, $\sigma_A$ decreases only by a factor of about 1.6, so the ratio of their logarithms

is about 1/5, i.e., $\sigma_A \propto \bar{\epsilon}^{-1/5}$. By contrast, $\sigma_A$ changes by a factor of about 5 as $\mathrm{Re}_\lambda$ is increased by a factor of nearly 3, so $\sigma_A \propto \mathrm{Re}_\lambda^{3/2}$. This quantifies the high sensitivity of $\sigma_A$ to changes of $\mathrm{Re}_\lambda$ compared to $\bar{\epsilon}$.

Two comments are here in order. First, we emphasize that we observe here $\sigma_A \propto \mathrm{Re}_\lambda^{3/2}$ instead of $\sigma_A \propto \mathrm{Re}_\lambda$. Therefore, there could be a critical $\mathrm{Re}_\lambda$, beyond which $\sigma_A \propto \mathrm{Re}_\lambda$ and below which $\sigma_A \propto \mathrm{Re}_\lambda^{3/2}$. However, the highest $\mathrm{Re}_\lambda$ in our DNS is 130. To verify this proposal, a large parameter range of $\mathrm{Re}_\lambda$ is required. Second, we note that $\sigma_A \propto \bar{\epsilon}^{-1/5}$. This is because

the Damköhler number increases with decreasing $\bar{\epsilon}$ (see Table 2), which is defined as the ratio of the fluid time scale to the characteristic thermodynamic time scale associated with the evaporation process $\mathrm{Da} = \tau_L/\tau_s$. Vaillancourt et al. (2002) also found that $\sigma_A$ decreases with $\bar{\epsilon}$, even though the mean updraft cooling is included in their study.

## 5   Discussion and conclusion

Condensational growth of cloud droplets due to supersaturation fluctuations is investigated using DNS. Cloud droplets are

tracked in a Lagrangian framework, where the momentum equation for inertial particles are solved. The thermodynamic equations governing the supersaturation field are solved simultaneously. Feedback from cloud droplets onto $\boldsymbol{u}$, $T$, and $q_v$ is included through the condensation rate and buoyancy force. We resolve the smallest scale of turbulence in all simulations. Contrary to the classical condensation theory, which leads to a narrow distribution when supersaturation fluctuations are ignored, we find that droplet size distributions broaden due to supersaturation fluctuations. For the first time, we explicitly demonstrate that

the size distribution becomes wider with increasing $\mathrm{Re}_\lambda$, which is, however, insensitive to $\bar{\epsilon}$. Supersaturation fluctuations are subjected to both temperature fluctuations and water vapor mixing ratio fluctuations.

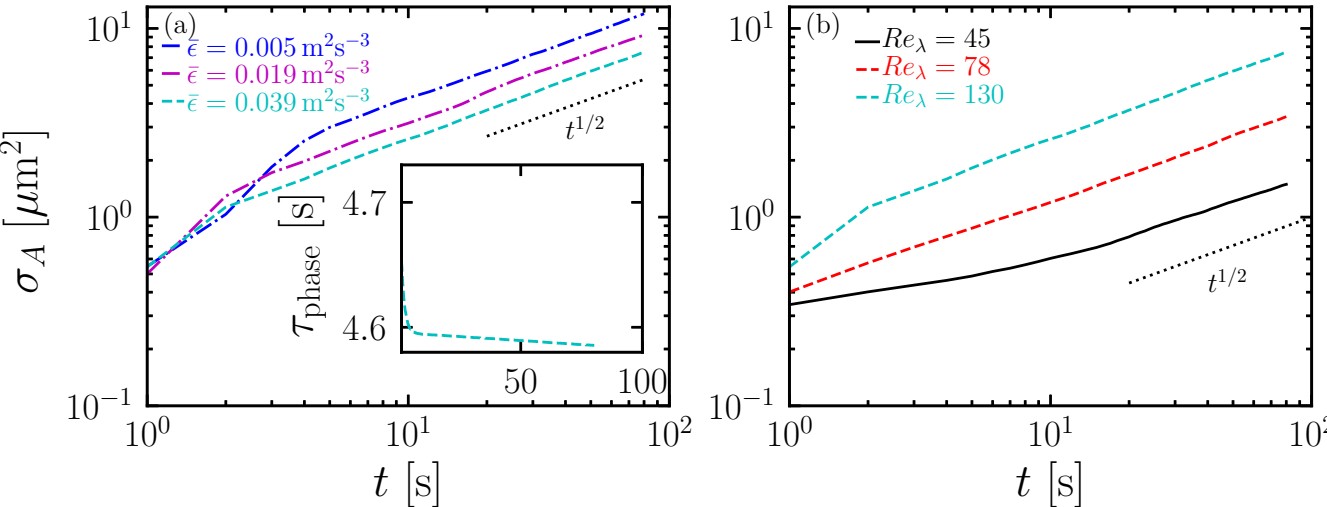

**Figure 4.** Time evolution of $\sigma_A$ for different (a) $\bar{\epsilon}$ at $\mathrm{Re}_\lambda = 130$ and (b) $\mathrm{Re}_\lambda$ at $\bar{\epsilon} = 0.039\,\mathrm{m^2 s^{-3}}$. Same simulations as in Figure 1.

We observe that $\sigma_A \propto \sqrt{t}$ when the complete sets of the thermodynamics equations governing the supersaturation are solved, which are consistent with the findings by Sardina et al. (2015) and Siewert et al. (2017). Even though fluctuations of temperature and water vapor mixing ratio, buoyancy force, and droplets feedbacks to the field quantities are neglected in their studies. This indicates that the stochastic model of condensational growth developed by Sardina et al. (2015) is robust. For the first

time, to our knowledge, the stochastic model (Sardina et al., 2015) and simulation results from the complete set of thermodynamics equations governing the supersaturation field are compared. The broadening size distribution with increasing $\mathrm{Re}_\lambda$ demonstrates that condensational growth due to supersaturation fluctuations is an important mechanism for droplet growth. The maximum $\mathrm{Re}_\lambda$ in the present study is 130, which is about two orders of magnitude smaller than the one in atmospheric clouds ($\mathrm{Re}_\lambda = 10^4$). Since the width of the size distribution increases dramatically with increasing $\mathrm{Re}_\lambda$, the supersaturation

fluctuation facilitated condensation may easily overcome the bottleneck barrier (Grabowski and Wang, 2013).

The stochastic model developed by Sardina et al. (2015) assumes that the width of droplet size distributions is independent of $\bar{\epsilon}$. Our result shows that the width decreases slightly with increasing $\bar{\epsilon}$. However, the largest $\bar{\epsilon}$ in warm clouds is about $10^{-3}\,\mathrm{m^2 s^{-3}}$ (Grabowski and Wang, 2013). Therefore, neglecting the smallest scales in the stochastic model is indeed acceptable. Vaillancourt et al. (2002) also found that the width of the droplet size distribution decreases with increasing $\bar{\epsilon}$

and attributed this to the decorrelation between supersaturation fluctuations and surface area of droplets. Sardina et al. (2015), however, found stronger correlation between supersaturation fluctuations and surface area of droplets with increasing $\mathrm{Re}_\lambda$. The present study is consistent with both the works of Vaillancourt et al. (2002) and Sardina et al. (2015). Therefore, we emphasize that there is *no* contradiction between both papers.

In the present study, the simulation box is stationary, which means that the volume is not exposed to cooling, as no mean updraft is considered. Therefore, the condensational growth is solely driven by supersaturation fluctuations. This is similar to the condensational growth of cloud droplets in stratiform clouds, where the updraft velocity of the parcel is close to zero (Hudson and Svensson, 1995; Korolev, 1995). The observational data shows that the width of the size distribution is wider than the one expected from condensational growth with a mean supersaturation (Hudson and Svensson, 1995; Brenguier et al., 1998; Miles et al., 2000; Pawlowska et al., 2006; Siebert and Shaw, 2017a). Qualitatively consistent with observations, we show that the width of droplet size distributions broadens due to supersaturation fluctuations.

Entrainment of dry air is not considered here. It may lead to rapid changes of the supersaturation fluctuations and result in an even faster broadening of the size distribution (Kumar et al., 2014). Activation of aerosols in a turbulent environment is omitted. This may provide a more physical and realistic initial distribution of cloud droplets. Incorporating all the cloud microphysical processes is computationally demanding, and will have be explored in future studies.

*Acknowledgements.* We thank Wojtek Grabowski, Andrew Heymsfield, Gaetano Sardina, Igor Rogachevskii and Dhrubaditya Mitra for stimulating discussions. This work was supported through the FRINATEK grant 231444 under the Research Council of Norway, SeRC, the Swedish Research Council grants 2012-5797 and 2013-03992, the University of Colorado through its support of the George Ellery Hale visiting faculty appointment, and the grant "Bottlenecks for particle growth in turbulent aerosols" from the Knut and Alice Wallenberg Foundation, Dnr. KAW 2014.0048. The simulations were performed using resources provided by the Swedish National Infrastructure for Computing (SNIC) at the Royal Institute of Technology in Stockholm and Chalmers Centre for Computational Science and Engineering (C3SE). This work also benefited from computer resources made available through the Norwegian NOTUR program, under award NN9405K. The source code used for the simulations of this study, the PENCIL CODE, is freely available on https://github.com/pencil-code/.

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
