# Peer review of "Cloud-droplet growth due to supersaturation fluctuations in stratiform clouds"

_Atmospheric Chemistry and Physics, 2018_

## Referee Comment (RC1) · Anonymous Referee #1 · 11 Oct 2018

General comments:

This paper presents a DNS study on droplet growth by condensation in turbulence. The purpose of this study is to explore the influence of supersaturation fluctuation on the broadening of droplet size distribution and to investigate the Reynolds number dependency of the broadening. The microphysics is solved by using the Lagrangian superdroplet method. By comparing the numerical results from the condensation at different Reynolds numbers and dissipation rates, the authors concluded that the supersaturation fluctuations produce broader droplet size distributions. The manuscript represents a good contribution to the development of new theories for the condensation process and is of potential interest for Atmospheric Chemistry and Physics community. However, by careful reading, some arguments in the context may seem hand-waving

and are not sufficiently robust to derive the main conclusion, and the evidence that the authors have cited are not firmly supportive. I would suggest that the authors provide more physical explanations and plots for the arguments. I would support the publication of this paper after the authors consider carefully the comments listed below.

Specific comments:

-Page 6/line 16: My main and critical points to the employed numerical framework is the choice of the timestep. It is not true that the Kolmogorov time scale is the smallest of the system. For 10 micrometres droplet, the particle response time defined in equation (15) is several order of magnitude lower. Unphysical droplet trajectories can be generated used such a large time step. Which temporal integration scheme is employed to solve equation (14)? Saito Gotoh used an implicit scheme and nevertheless their time step is much smaller than the Kolmogorov time scale. Can the authors comment on this issue? A validation case must be provided (at least for one of the low-resolution cases) with a much smaller time step. If the results differ, an entirely new dataset must be generated for the paper.

-If the time step is the Kolmogorov time scale, why is the maximum time of simulation limited to 80 s? The maximum number of iterations will be 4000 that is not so difficult to reach in a supercomputer with few hours of computational time.

-Why do you evolve superparticles? Can the authors not evolve the actual number of particles inside the domain? The maximum number of droplets that need to be evolved is about 30 million that again is not so prohibitive in a modern supercomputer. State of the art of droplet-laden DNS has reached much higher droplet numbers. -Connecting the previous points: How long computational time is needed for the smaller and the larger case? How many cores have you used?

-Page 1/l. 7-8: the adverbs "strongly" and "weakly" (which also appear in other parts of the manuscript) are not fully supported by the results provided in the paper. I can see differences below one order of magnitude smaller between the lines in the plots (e.g.

Fig. 4). The range of Reynolds number is quite limited to appreciate "strongly" and "weakly" variations. The authors can modify the random forcing term to achieve higher Reynolds.

-Page 1/l. 11: the simulations have been done without updraft. The authors should add a paragraph in the introduction of the effects and consequences of the updraft in the broadening of droplet size distributions.

-Page 2/l. 17: Paoli Sharif results are strongly influenced by an arbitrary forcing term for the temperature and water vapor equations

-Page 2/l. 26-27 (and many other locations in the manuscript): Can the authors comment on the sentence "solve the thermodynamics" when the maximum temperature fluctuations of their system are 0.1 K?

-Page 3/l. 8: Can the authors provide a plot with the ratio between $a_{rms}$ and $B_{rms}$ where a is the fluid acceleration (the material derivative of the velocity)? My feeling is that at these small scales buoyancy effects can be neglected.

-Page 3/l. 31: A theoretical issue: the velocity field within the Boussinesq approximation is divergence free that is not the case. A short paragraph should be added to justify this theoretical mismatch briefly.

-Page 8/l. 5: How can you see that equation 18 follows a Brownian motion?

-Page 11/l.15: "Therefore, neglecting the smallest scales in the stochastic model is indeed acceptable", the stochastic models are derived under the hypothesis of large-scale separation so that they cannot be applied at $Re_\lambda = 40$. If you want to show slightly less dependence repeat the same simulation set up with three different dissipation for the higher resolution setup.

-Page 12/l.3: I guess that the contradiction is due to the presence of updraft

-The three appendices containing just one definition are not needed, please move in

the main text

Technical corrections:

-Pag3 3/l.13: is→ are

-Page 3/l.22: there is a 0 after the citation Li et al, 2017

-Page 3/l.29: provide a reference for the code

-Page 4/l.11: index and vectorial notations should not be mixed

-Page 6/l.5: is the nonlinear correction needed? What is the range of droplet Reynolds number?

-Page 6/l.25: I guess the factor $2^\beta$ is wrong, otherwise, it would be $2^{64}$ for the larger case!!!

-Page 6: there is no need to create a new subsection 3.2

-Page 7/l.7: fix KOlmogorov

---

## Referee Comment (RC2) · Anonymous Referee #2 · 16 Oct 2018

"Cloud-droplet growth due to supersaturation fluctuations in stratiform clouds"

by Xiang-Yu Li, Gunilla Svensson, Axel Brandenburg, and Nils E. L. Haugen

The focus of this paper is the influence of supersaturation fluctuations on droplet condensation growth, which has become an active area of research in recent years. To have the stratiform clouds as a motivation, authors have studied this effect in the absence of the mean updraft velocity. In this study, the conservation of momentum and scalar (temperature and water vapor) equations are solved using the direct numerical simulation (DNS) in a rectangular domain and the random velocity forcing drives the turbulence. Here, the Eulerian scalar and momentum field is coupled with the Lagrangian droplet dynamics using the superparticle method. Additionally, the physics of droplet activation and droplet collision-coalescence process were ignored. All droplets

were considered at an initial size of 10 um and the starting supersaturation in the domain was 2%. Authors have examined cases of different Taylor Reynolds number ($Re_\lambda$) and mean kinetic energy dissipation rate ($\epsilon$).

In general, the approach here is very much similar to that of Sardina et al. (2015), Siewert at al. (2017) and others. The only significant difference is the treatment of supersaturation field; in the current case, it is obtained by solving temperature and water vapor conservation equations contrary to the assumption of supersaturation field as a passive scalar in previous studies. Moreover, the authors compared the results with the stochastic formulation of Sardina et al. (2015) and other numerical-simulation studies. The results are consistent with the other studies, the droplet size dispersion ($\sigma_A$) growth is proportional to $t^{1/2}$. Similarly, the broadening in droplet size distribution is shown to be nearly independent of $\epsilon$ (a slight decrease), however, it increases with increase in $Re_\lambda$ consistent with the conclusions of Sardina et al. (2015).

Review points:

- The authors should be clear about the novelty. The main significant differences between current simulation and previous are the treatment of supersaturation field and the feedback due to condensation. Although, authors also acknowledge that the treatment of supersaturation as a passive scalar is sufficient. Furthermore, they explicitly showed that the results are independent to the dissipation rate ($\epsilon$) which was not clearly presented in the other studies. Please update abstract, intro and conclusions to make clear.

- Claimed relevance is to stratocumulus clouds, but entrainment of unsaturated air and possible secondary activation is know to strongly change droplet size distribution in that system. How does absence of entrainment limit the results presented? What changes can be expected when entrainment and activation are included? These limitations should be discussed.

- Page-6, Line 16: It should be supersaturation instead of saturation.

- Page-7, Line 7: Fix the typo

- The assumption used to get the eq. 20 is not required to derive the equation for $\sigma$A growth.

- The phase relaxation time might be changing with time due to the mean radius growth (specifically, at the starting since there is a starting supersaturation around 2%). It might cause some deviation in the result ($\sigma_A$ vs $t$) from the $t^{1/2}$ relation. Authors should discuss this effect along with the discussion of figure 4.
* * *

---

## Author Comment (AC1) · 4 Nov 2018

> General comments: > This paper presents a DNS study on droplet growth by condensation in > turbulence. The purpose of this study is to explore the influence > of supersaturation fluctuation on the broadening of droplet size > distribution and to investigate the Reynolds number dependency of > the broadening. The microphysics is solved by using the Lagrangian > superdroplet method. By comparing the numerical results from the > condensation at different Reynolds numbers and dissipation rates, the > authors concluded that the supersaturation fluctuations produce broader > droplet size distributions. The manuscript represents a good contribution > to the

development of new theories for the condensation process and is > of potential interest for Atmospheric Chemistry and Physics community. > However, by careful reading, some arguments in the context may seem > hand-waving and are not sufficiently robust to derive the main conclusion, > and the evidence that the authors have cited are not firmly supportive. I > would suggest that the authors provide more physical explanations and > plots for the arguments. I would support the publication of this paper > after the authors consider carefully the comments listed below.

We thank the reviewer for his/her constructive remarks. As explained below in detail, we have now tried to make our arguments more robust. All our changes are highlighted in blue.

Specific comments: > -Page 6/line 16: My main and critical points to the employed numerical > framework is the choice of the timestep. It is not true that the > Kolmogorov time scale is the smallest of the system. For 10 micrometres > droplet, the particle response time defined in equation (15) is > several order of magnitude lower. Unphysical droplet trajectories can be > generated used such a large time step. Which temporal integration scheme > is employed to solve equation (14)? Saito Gotoh used an implicit scheme > and nevertheless their time step is much smaller than the Kolmogorov > time scale. Can the authors comment on this issue? A validation case > must be provided (at least for one of the low-resolution cases) with a > much smaller time step. If the results differ, an entirely new dataset > must be generated for the paper.

The smallest physical time step in the system is indeed the particle response time. This was always handled correctly in the code, but we didn't describe this correctly. We have now corrected the text in the first paragraph of section 3.1 on page 7. A validation is provided in the response as supplementary material. Since our simulation time step is smaller than the smallest physical time step, a shorter time step gives identical results.

> -If the time step is the Kolmogorov time scale, why is the maximum > time of simulation limited to 80 s? The maximum number of iterations > will be 4000 that is not so

difficult to reach in a supercomputer with > few hours of computational time.

As explained above, the smallest time step in the system is indeed the particle response time. We have now corrected the text in the first paragraph of section 3.1 on page 7. A validation is provided in the response.

> -Why do you evolve superparticles? Can the authors not evolve the actual > number of particles inside the domain? The maximum number of droplets that > need to be evolved is about 30 million that again is not so prohibitive > in a modern supercomputer. State of the art of droplet-laden DNS has > reached much higher droplet numbers. -Connecting the previous points: > How long computational time is needed for the smaller and the larger > case? How many cores have you used?

We emphasized in lines 22-26 on page 3 that "For condensational growth, the super-particle approach (Li et al., 2017) is the same as the Lagrangian point-particle approach (Kumar et al., 2014) since there is no interactions among droplets. Nevertheless, we still use the superparticle approach so that we can include more processes like collection (Li et al., 2017, 2018) in future. Another reason to adopt superparticle approach is that it can be easily adapted to conduct Large-eddy simulations with appropriate sub-grid scale models (Grabowski and Abade, 2017)."

We have now added the description of the CPU cost at the end of section 3.2 on page 7.

> -Page 1/l. 7-8: the adverbs "strongly" and "weakly" (which also > appear in other parts of the manuscript) are not fully supported by the > results provided in the paper. I can see differences below one order > of magnitude smaller between the lines in the plots (e.g. Fig. 4). The > range of Reynolds number is quite limited to appreciate "strongly" > and "weakly" variations. The authors can modify the random forcing > term to achieve higher Reynolds.

We have now replaced "strongly" and "weakly" in the various places in the manuscript

by more precise statements by writing: that \sigma_A is proportional to Re_lambda to the 3/2 power, but only proportional to \bar\epsilon to the -1/5 power.

>- -Page 1/l. 11: the simulations have been done without updraft. The authors should > add a paragraph in the introduction of the effects and consequences of the updraft in > the broadening of droplet size distributions.

We have now added the following discussion at the end of the second paragraph of section 1 on page 2: "When the mean updraft velocity is not zero, there could be a competition between the mean updraft velocity and supersaturation fluctuations. This may diminish the role of supersaturation fluctuations (Sardina et al. 2018)."

> -Page 2/l. 17: Paoli Sharif results are strongly influenced by an > arbitrary forcing term for the temperature and water vapor equations

We have now addressed the "arbitrary forcing term" by saying "turbulence as well as stochastically forced temperature and vapor".

> -Page 2/l. 26-27 (and many other locations in the manuscript): Can the > authors comment on the sentence "solve the thermodynamics" when the > maximum temperature fluctuations of their system are 0.1 K?

The main difference between the present study and that of Sardina et al. is that we solve the supersaturation explicitly by solving the temperature and water vapor mixing ratio field. This is why we say that we "solve for the temperature field". Due to the limited Reynolds number in DNS, the temperature fluctuations are small. In this sense, it is indeed acceptable to treat the supersaturation as a passive scalar. Nevertheless, solving for the temperature and water vapor mixing ratio directly is beneficial when one wants to incorporate entrainment and so on, which is an ongoing project.

> -Page 3/l. 8: Can the authors provide a plot with the ratio between a rms and B rms > where a is the fluid acceleration (the material derivative of the velocity)? My feeling is > that at these small scales buoyancy effects can be neglected.

The forced turbulence becomes stationary. In this case, a_rms/B_rms=250. Therefore, the buoyancy force is indeed small. We have now discussed this in the last paragraph of page 8. The figure showing a_rms/B_rms is attached as supplementary material.

> -Page 3/l. 31: A theoretical issue: the velocity field within the > Boussinesq approximation is divergence free that is not the case. A short > paragraph should be added to justify this theoretical mismatch briefly.

We discussed that "The background air flow is almost incompressible and thus obeys the Boussinesq approximation." in the first paragraph of section 2.1.

> -Page 8/l. 5: How can you see that equation 18 follows a Brownian motion?

We have now changed it to the following, "It can be seen from \Eq{eq:A} that the evolution of the surface area is analogy to Brownian motion," at P.10/l.3.

> -Page 11/l.15: "Therefore, neglecting the smallest scales in the stochastic model is > indeed acceptable", the stochastic models are derived under the hypothesis of large- > scale separation so that they cannot be applied at Re $\lambda$ = 40. If you want to show slightly > less dependence repeat the same simulation set up with three different dissipation for > the higher resolution setup.

We have now updated all the figures for different \epsilon with Re_lambmda=130.

> -Page 12/l.3: I guess that the contradiction is due to the presence of updraft

We have now added the following discussion at P.14/l.1-2, "It could also be due to the mean updraft cooling included in the model of Vaillancourt et al. (2002), which was excluded in the present study and in the work of others."

> -The three appendices containing just one definition are not needed, please move in > main text

We have now moved all the appendix to the main text.

> Technical corrections: > -Pag3 3/l.13: is→ are

We have now corrected it.

> -Page 3/l.22: there is a 0 after the citation Li et al, 2017

We have now corrected it.

> -Page 3/l.29: provide a reference for the code

We provided it in the last sentence of the acknowledgement.

> -Page 4/l.11: index and vectorial notations should not be mixed

We have now only adopted the tensor notation.

> -Page 6/l.5: is the nonlinear correction needed? What is the range of droplet Reynolds > number?

It is between 3 to 5.5, which is almost negligible. Nevertheless, we always turn this on so that collision-coalescence can be investigated as a sequential work.

> -Page 6/l.25: I guess the factor 2 $\beta$ is wrong, otherwise, it would be 2 64 for the larger > case!!!

We have now removed the statement because $\beta$ is not used elsewhere in the manuscript.

> -Page 6: there is no need to create a new subsection 3.2

Section 3.2 is the DNS, which is to be distinguished from the initial configuration in 3.1.

> -Page 7/l.7: fix Kolmogorov

We have now fixed it,
* * *
[Figure]

[Figure]

**Fig. 1.** A validation of the time step

[Figure]

**Fig. 2.** Plot of a_rms/B_rms

---

## Author Comment (AC2) · 4 Nov 2018

> The focus of this paper is the influence of supersaturation fluctuations > on droplet condensation growth, which has become an active area of > research in recent years. To have the stratiform clouds as a motivation, > authors have studied this effect in the absence of the mean updraft > velocity. In this study, the conservation of momentum and scalar > (temperature and water vapor) equations are solved using the direct > numerical simulation (DNS) in a rectangular domain and the random > velocity forcing drives the turbulence. Here, the Eulerian scalar and > momentum field is coupled with the Lagrangian droplet dynamics using the > superparticle method. Additionally, the

physics of droplet activation > and droplet collision-coalescence process were ignored. All droplets were > considered at an initial size of 10 um and the starting supersaturation > in the domain was 2%. Authors have examined cases of different Taylor > Reynolds number (Re $\lambda$ ) and mean kinetic energy dissipation rate (). > In general, the approach here is very much similar to that of Sardina > et al. (2015), Siewert at al. (2017) and others. The only significant > difference is the treatment of supersaturation field; in the current > case, it is obtained by solving temperature and water vapor conservation > equations contrary to the assumption of supersaturation field as a > passive scalar in previous studies. Moreover, the authors compared the > results with the stochastic formulation of Sardina et al. (2015) and other > numerical-simulation studies. The results are consistent with the other > studies, the droplet size dispersion ($\sigma$ A) growth is proportional to > tˆ1/2. Similarly, the broadening in droplet size distribution is shown > to be nearly independent of (a slight decrease), however, it increases > with increase in Re $\lambda$ consistent with the conclusions of Sardina et > al. (2015).

We thank the reviewer for his/her constructive comments and have now emphasized the novelty of our work in various places. Our detailed response to the reviewer's comments are explained below, highlighted in blue.

> Review points: > - The authors should be clear about the novelty. The main significant > differences between current simulation and previous are the treatment of > supersaturation field and the feedback due to condensation. Although, > authors also acknowledge that the treatment of supersaturation as a > passive scalar is sufficient. Furthermore, they explicitly showed that > the results are independent to the dissipation rate () which was not > clearly presented in the other studies. Please update abstract, intro > and conclusions to make clear.

We have now added the following in: 1. abstract "The supersaturation field is calculated directly by simulating the temperature and water vapor fields instead of treating it as a passive scalar. Thermodynamic feedbacks to the fields due to condensation are also included."

"Also, for the first time, we explicitly demonstrate that the time evolution of the size distribution..."

2. introduction We addressed that P.2/l.29-30: "Neither Sardina et al. (2015) nor Siewert et al. (2017) solved the thermodynamics that determine the supersaturation field." P.3/l.11-12: "where turbulence, thermodynamics, feedback from droplets to the fields via the condensation rate and buoyancy force are all included." P.3/l.15-17: "For the first time, to our knowledge, the stochastic model and simulation results from the complete set of equations governing the supersaturation field is compared."

3. conclusion P13/l5-6: "For the first time, we explicitly demonstrate that the size distribution becomes wider with increasing Re_lambda, which is, however, insensitive to \bar\epsilon."

> - Claimed relevance is to stratocumulus clouds, but entrainment of unsaturated air and > possible secondary activation is know to strongly change droplet size distribution in that > system. How does absence of entrainment limit the results presented? What changes > can be expected when entrainment and activation are included? These limitations > should be discussed.

We have now added the following: P.14/10-13: "Entrainment of dry air is not considered here, which may lead to very rapid changes of supersaturation fluctuations and result in fast broadening of the size distribution Kumar et al. (2014). Activation of aerosols in a turbulent environment is omitted. This may provide a more physical and realistic initial distribution of cloud droplets. Incorporating all the cloud microphysical processes is computationally challenging, which will be explored further in the future studies."

> - Page-6, Line 16: It should be supersaturation instead of saturation.

We have now corrected it.

> - Page-7, Line 7: Fix the typo

We have now fixed it.

[Figure]

> - The assumption used to get the eq. 20 is not required to derive the equation for $\sigma A$ growth.

The scaling law \sigma_A~tˆ{1/2} does not require the assumption. However, to obtain eq.23, T_0»\tau_phase is needed. We have now added the explanation below eq 22 on page 10.

> - The phase relaxation time might be changing with time due to the mean radius growth > (specifically, at the starting since there is a starting supersaturation around 2%). It > might cause some deviation in the result ($\sigma$ A vs t) from the t 1/2 relation. Authors should > discuss this effect along with the discussion of figure 4.

We have now added a discussion in the last paragraph at P.12/l.9-12.

---

## Author Response (AR2)

Report#2
>I have read through the responses to reviewer comments and the revised paper,
>and am satisfied that the main points have been addressed. I do not intend
>to delay the publication of the paper, but on the second reading I notice
>that some important references are missing and should be added to the
>introduction so that connections to the prior and current literature can be identified.
>The paper by W. A. Cooper (J. Atmos. Sci. 1989 "Effects of variable droplet growth
>histories on droplet size distributions. Part I: Theory") should be cited as one
>of the key steps in identifying supersaturation fluctuations as relevant to cloud
>droplet size distribution.

We have now cited the paper of W. A. Cooper as follows (see lines 19-22 on page 2),
"Cooper (1989) proposed that droplets moving in clouds are exposed to a varying supersaturation field. This results in broadening of droplets size distribution due to supersaturation fluctuations. Grabowski and Wang (2013) called the mechanism of Cooper (1989) the eddy-hopping mechanism, which was then investigated by Grabowski and Abade (2017)"

>There are many papers from the Russian literature on
>stochastic condensation, going back to 1970s, so at least some representative
>ones should be mentioned (e.g. Mazin, Sedunov, and more recent Khvorostyanov).

We have now cited literature of Mazin, Sedunov, and more recently Khvorostyanov from l12-13 on page 2.

>Some recent lab studies also exist (e.g. Desai et al 2018 JAS).

We have now cited Desai et al 2018 JAS in line 6 on page 3.

List of changes:

1. Line 13-14 on page 2

"focus (Sedunov, 1965; Kabanov and Mazin, 1970; Cooper, 1989; Srivastava, 1989; Korolev, 1995; Khvorostyanov and Curry, 1999; Sardina et al., 2015; Grabowski and Abade, 2017)."

2. Line 19-23 on page 2

"Cooper (1989) proposed that droplets moving in clouds are exposed to a varying supersaturation field. This results in broadening of droplets size distribution due to supersaturation fluctuations. Grabowski and Wang (2013) called the mechanism of Cooper (1989) the eddy-hopping mechanism, which was then investigated by Grabowski and Abade (2017)."

"the mean energy dissipation rate of"

3. Line 26 on page 2

"due to turbulence mixing"

4. Line 6 on page 3

"and Desai et al. (2018)"

5. Line 14-18 on page 13

[revised manuscript text omitted]